# Microstructure and Mechanical Properties of Alumina Composites with Addition of Structurally Modified 2D Ti_3_C_2_ (MXene) Phase

**DOI:** 10.3390/ma14040829

**Published:** 2021-02-10

**Authors:** Tomasz Cygan, Jaroslaw Wozniak, Mateusz Petrus, Artur Lachowski, Wojciech Pawlak, Bogusława Adamczyk-Cieślak, Agnieszka Jastrzębska, Anita Rozmysłowska-Wojciechowska, Tomasz Wojciechowski, Wanda Ziemkowska, Andrzej Olszyna

**Affiliations:** 1Faculty of Material Science and Engineering, Warsaw University of Technology, Woloska 141 St, 02-507 Warsaw, Poland; jaroslaw.wozniak@pw.edu.pl (J.W.); mateusz.petrus.dokt@pw.edu.pl (M.P.); boguslawa.cieslak@pw.edu.pl (B.A.-C.); agnieszka.jastrzebska@pw.edu.pl (A.J.); anita.wojciechowska.dokt@pw.edu.pl (A.R.-W.); andrzej.olszyna@pw.edu.pl (A.O.); 2Institute of High Pressure Physics, Polish Academy of Sciences, Sokolowska 29/37 St, 01-142 Warsaw, Poland; artur@unipress.waw.pl; 3Faculty of Mechanical Engineering, Lodz University of Technology, Stefanowskiego 1/15 St, 90-924 Lodz, Poland; wojciech.pawlak@p.lodz.pl; 4Faculty of Chemistry, Warsaw University of Technology, Noakowskiego 3 St, 00-664 Warsaw, Poland; wojciechowski16@gmail.com (T.W.); ziemk@ch.pw.edu.pl (W.Z.)

**Keywords:** Al_2_O_3_, Ceramic, Ti_3_C_2_, MXene, ceramic matrix composites, spark plasma sintering, microstructure, mechanical properties

## Abstract

This study presents new findings related to the incorporation of MXene phases into ceramic. Aluminium oxide and synthesised Ti_3_C_2_ were utilised as starting materials. Knowing the tendency of MXenes to oxidation and degradation, particularly at higher temperatures, structural modifications were proposed. They consisted of creating the metallic layer on the Ti_3_C_2_, by sputtering the titanium or molybdenum. To prepare the composites, powder metallurgy and spark plasma sintering (SPS) techniques were adopted. In order to evaluate the effectiveness of the applied modifications, the emphasis of the research was placed on microstructural analysis. In addition, the mechanical properties of the obtained sinters were examined. Observations revealed significant changes in the MXenes degradation process, from porous areas with TiC particles (for unmodified Ti_3_C_2_), to in situ creation of graphitic carbon (in the case of Ti_3_C_2_-Ti/Mo). Moreover, the fracture changed from purely intergranular to cracking with high participation of transgranular mode, analogously. In addition, the results obtained showed an improvement in the mechanical properties for composites with Ti/Mo modifications (an increase of 10% and 15% in hardness and fracture toughness respectively, for specimens with 0.5 wt.% Ti_3_C_2_-Mo). For unmodified Ti_3_C_2_, enormously cracked areas with spatters emerged during tests, making the measurements impossible to perform.

## 1. Introduction

Due to their attractive properties, advanced ceramics are used as structural materials in many industrial sectors. They are characterised by high hardness and high compressive strength, good wear resistance, chemical inertness and stability of mechanical properties at elevated temperatures. These properties make them perfect candidates for the parts working in such conditions as: corrosive environments, high temperature, under high pressure, or in contact with abrasive particles. What limits their broader use, however, is low tensile strength, mechanical unreliability, susceptibility to thermal shocks, and their biggest disadvantage, brittleness [1,2,3,4,5,6,7,8,9]. To overcome these drawbacks (fully or at least partially), ceramic matrix composites (CMC) have been developed. Depending on the desired results, the most commonly used reinforcements are either ceramic (improving hardness, fracture toughness, bending strength, heat resistance) or metallic (increasing fracture toughness, bending strength, electrical and thermal conductivity). These phases can take many different forms, from isometric particles, through long fibres and whiskers, to high surface flakes [10,11,12,13,14,15]. In particular, since the discovery of graphene, the most promising materials suitable for reinforcing are precisely 2D structures. They possess extraordinary properties which emerge directly from their unique morphology. There are several known materials that are characterised by a two-dimensional layout, e.g.,: the aforementioned graphene and its derivatives, molybdenum disulfide, tungsten ditelluride, hexagonal boron nitride or phosphorene. Recent progress made within the field of materials science resulted in other structures joining this list, the MXenes [16,17,18,19,20,21,22,23,24,25].

MXenes derive directly from M_n+1_AX_n_ materials (known as MAX phases), where M = early transition metal (e.g., Ti, Mo, V, Nb, Cr), A is an element from group 13 or 14, X = carbon or nitrogen, and n = 1–3. They are synthesized by selective etching of the A layer in aqueous fluoride-containing acids. Such treatment results in terminating the surface with several functional groups (oxygen, hydroxyl or fluoride), and so the obtained MXenes can be described with M_n+1_X_n_T_x_ composition, where T_x_ included in the formula refers to these functional groups. In the last step of the process, etched precipitates are mechanically exfoliated by sonification, which leads to the two-dimensional structures being obtained [26,27,28,29,30]. As in the case of graphene and other 2D materials, the characteristic morphology, but also the surface functional groups, causes the MXene phases to possess many interesting properties. They exhibit high electrical and thermal conductivity, high Young modulus, excellent electrochemical activity, hydrophilic nature, tuneable bandgap, capability to intercalate ions and high electrical capacity. Such versatility of MXenes makes them attractive materials for many applications, e.g., energy storage and conversion, electronic devices, environmental and biomedical parts, catalysis or sensors [31,32,33,34,35,36,37,38,39,40]. However, they are susceptible to oxidation, which starts at 200–250 °C and 750–800 °C for air and argon atmosphere, respectively, and leads to full degradation above 1000 °C in either case. This limits their broader application and brings technological challenges to the process of producing composites with the addition of MXene phases. In particular, in the case of ceramic and metallic matrices, this requires significantly higher consolidation temperatures compared to polymers [41,42,43].

To date, there have only been isolated articles in literature regarding ceramic matrix composites with MXene phases. The first published work concerned Al_2_O_3_ reinforced with Ti_3_C_2_ crystals, and focused mainly on the mechanical properties of composites [44]. It is important to note however, that phases used in this research were not subjected to sonification and consequently not delaminated after etching. In addition, the authors report a significant increase in fracture toughness, bending strength and hardness, although it was an outcome in comparison to the reference sample which was not fully densified (and characterised by poor mechanical properties). In another article, the same MXene phase was incorporated into the ZnO matrix, and the emphasis of the research was placed on investigating mechanical and electrical properties [45]. To prevent thermal degradation of the reinforcement, composites were consolidated using the cold sintering process, which is carried out below 300 °C. The authors report accomplishing high density, significant improvement in electrical conductivity, as well as considerable hardness and an increase to elastic modulus. In subsequent publication on the subject, the microstructure and mechanical properties of SiC-Ti_2_C composites were analysed [46]. The spark plasma sintering (SPS) method was utilized for specimens sintering, which achieved close to theoretical densities. In addition, authors report results indicating an increase of hardness and fracture toughness, with crack deflection and bridging mechanisms responsible for reinforcement. In the most recent study, Ti_3_C_2_ was incorporated into silicon nitride ceramic, and its influence on the microstructure and mechanical properties was analysed [47]. However, as pointed out by the authors, the utilised oxide sintering activators resulted in full degradation of the MXene during the SPS process. Interestingly, it limited the matrix phase transformation and subsequently created additional Si_2_N_2_O phase. An increase in mechanical properties was also observed and, in particular, a significant improvement in the hardness has been achieved.

In the present study, alumina matrix composites reinforced with Ti_3_C_2_ were prepared with the use of powder metallurgy and consolidated using SPS. To overcome the MXenes tendency for oxidation during the sintering process, the structural modifications were proposed. They consisted of sputtering the titanium or molybdenum onto the Ti_3_C_2_ to create the metallic layer. Such alterations should prevent MXene from full degradation and additionally ensure a better interface between matrix and reinforcement. To our knowledge, this is the first experiment to introduce structurally modified 2D titanium carbide into the ceramic composites.

## 2. Materials and Methods

### 2.1. Substrates

The following commercially available powders were used in this work: titanium (GoodFellow Cambridge Ltd., Huntingdon, UK) with a chemical purity of 99.6% and the average particle size below 20 µm, aluminium (Benda-Lutz, Skawina, Poland) with a chemical purity of 99.7%, synthetic graphite (Sigma Aldrich, Poznań, Poland) with a chemical purity of 99.9% and the average particle size below 20 µm, and α-Al_2_O_3_ (Taimei Chemicals Co., Ltd., Tokyo, Japan) with a chemical purity of 99.99% and the average particle size of 135 nm.

### 2.2. Synthesis of The Ti_3_AlC_2_ MAX Phase

The starting point for obtaining MXene is the synthesis of MAX phase, which in this study was produced by powder metallurgy and SPS. Firstly, Ti:Al:C elements were wet blended in a ratio of 3:1:1.9, with the process carried out in isopropyl alcohol for 24 h with the use of a ball mill. The mixture obtained was dried, granulated, and then subjected to synthesis in SPS (HP D 10, FCT Systeme GmbH, Effelder-Rauenstein, Germany) with the use of specially designed moulds-stamps set, which allows a pressureless process to be followed [48]. The parameters were as follows: heating rate 250 °C/min, temperature 1300 °C, dwell time 3min, vacuum. Subsequently, the obtained MAX phase was ground with automatic mortar grinder (Retsch KM100) and sieved (# = 300 µm) with automatic sieve shaker (Haver EML 200 Premium).

### 2.3. Preparation of The Ti_3_C_2_ MXene Phase

Removal of the aluminium (Al) layer from the Ti_3_AlC_2_ powder, was carried out via standard procedure of concentrated hydrofluoric acid (HF) etching, which is a universal and widely accepted method for different MXenes preparation, as described in previous papers [49,50,51]. Briefly, MAX powder was slowly (in small portions) added to a continuously mixed (1000 rpm) 48% water solution of HF (Sigma Aldrich, Poznań, Poland), to finally obtain an exothermic reactive mixture of 1g MAX for 10 g of HF. The process was carried out for 24 h until all available Al layers and hydrogen (H_2_) vapours were removed. The powder was then decanted, collected and washed with deionized water until pH of the clay reached ca. 5–6. The obtained clay composed of Ti_3_C_2_T_x_ harmony-like typical MXene structure was dried at room temperature (RT), and stored at 5 °C in the dark for further processing steps.

The multilayered Ti_3_C_2_ flakes were obtained using the developed disintegration procedure, purposely designed to avoid delamination into single-layered flakes, that are known for their proneness to rapid oxidation and decomposition during the sintering process [52]. It was previously shown, that flakes with the multilayered structure are more stable compared to single-layered flakes. Therefore, it is reasonable to test such a structure in ceramic matrix composites.

The disintegration procedure consisted of two important steps of tip sonication, involving hexane and isopropanol solvents (Sigma-Aldrich) with different polarities. The process was performed in an argon atmosphere, with a ratio of 1 g powder to 50 cm^3^ of medium. In the first step, the sonication was carried out in dried non-polar hexane for 2 h, after which the sediment was collected and dried at room temperature (RT), and subsequently, subjected to sonication in dried polar isopropanol for 1 h. In both cases, the process was performed in periodical working mode (1 s/3 s on/off), and in an ice bath to avoid the overheating. After drying at RT, the Ti_3_C_2_T_x_ flakes were stored at 5 °C in the dark.

### 2.4. Structural Modification of Ti_3_C_2_ MXene Phase

In the next technological step, the obtained MXene phase was coated with a layer of molybdenum or titanium. For each of the processes, Ti_3_C_2_ powder of a mass of 0.5 g was placed in a rotary holder with SiO_2_ beads of 1mm diameter used as a mixing agent. The bottom surface of the holder was 30° tilted to the horizontal plane. After slow pumping to the 1000 Pa of pressure the diffusion pump was started. When the vacuum reached the level of 3 × 10^−3^ Pa, pure argon (5N) was introduced and the working pressure was set at a 1.50 Pa. The magnetron of 2″ diameter equipped with target of 99.9% was ignited with the power 500 W and 300 W, for Ti and Mo, respectively. The maximum temperature of 100 °C was reached after 1h. Deposition time was 29.2/25.5 ks, and sputtered volume of targets was 0.527/0.577 cm^3^, both for Ti and Mo, respectively. After the process the beads were sieved to collect Ti_3_C_2_-Ti and Ti_3_C_2_-Mo powders.

The main purpose of this modification was to prevent Ti_3_C_2_ from degradation during the sintering process. In addition, the metallic Ti/Mo layer provides a better interface between matrix and reinforcement, which translates into higher mechanical properties of composites. Such an approach was previously used for graphene modification, where nickel layer improved wettability between matrix and reinforcement [53]. However, to the best of our knowledge, there are no reports on the use of MXene phase modification approaches for application in ceramic matrix composites.

### 2.5. Al_2_O_3_-Ti_3_C_2_ Powders Preparation and Sintering

The composites presented in this study were prepared with the use of powder metallurgy and SPS. In the first technological step, the Al_2_O_3_-x wt.% Ti_3_C_2_/Ti_3_C_2_-Ti/Ti_3_C_2_-Mo mixtures (where x = 0.5, 1, 2), were wet blended in isopropanol for 8h with the use of attritor type mill and alumina balls as grinding agent. The slurries obtained were dried and sieved (# = 300 µm) with an automatic sieve shaker (Haver EML 200 Premium). Subsequently, the mixtures were placed in sets of graphitic moulds/punches and sintered with the use of SPS (HP D 10, FCT Systeme GmbH, Germany). The parameters of the process were as follows: sintering temperature 1400 °C, heating/cooling rate 250 °C/min, 3 min dwell time, 35 MPa applied pressure and vacuum. All of the obtained samples were then ground to remove remains of graphite after the sintering process. In addition, the unmodified sinter of pure alumina was prepared as a reference specimen. To ensure the required surface roughness and parallelism for further investigations, all of the surfaces were mechanically ground and polished with diamond solution to a grit size of 1 µm.

### 2.6. Ti_3_AlC_2_ MAX and Ti_3_C_2_/Ti_3_C_2_-Mo/Ti_3_C_2_-Ti MXene Phases Characterisation Methods

#### 2.6.1. Scanning Electron Microscopy (SEM) Observations

Scanning electron microscopy (SEM Hitachi S5500) with an EDS detector was used for powders examination at each technological step, i.e.,: synthesised Ti_3_AlC_2_, MAX phase after etching and delaminated Ti_3_C_2_ MXene phase.

#### 2.6.2. Fourier Transform Infrared Spectroscopy (FTIR) Measurements

To detect potential chemical bonding present in pristine Ti_3_C_2_ flakes and in MXenes modified with Ti/Mo, Fourier transform infrared spectroscopy (FTIR) was used (Nicolet iS5, Thermo Fisher Scientific, Waltham, MA, USA). The device was equipped with a diffuse reflectance infrared Fourier transform (DRIFT) accessory, and operated in the light wavelength range from 220 to 4000 cm^−1^. For measurement, each sample was mixed with dried KBr to obtain specimens with concentrations ca. 2.5 wt.% in KBr. The samples were then placed in a DRIFT cuvette and each spectrum was reported in the average of 30 scans. For data analysis, OMNIC (Thermo Fisher) software was used.

#### 2.6.3. Zeta Potential Measurements

Colloidal properties of the pristine Ti_3_C_2_ flakes and MXenes modified with Ti/Mo were performed in distilled water and in isopropanol with the use of NANO ZS ZEN3500 analyser (Malvern Instruments, Malvern, UK). The device is equipped with a back-scattered light detector operating at a 173° angle, and the studies were carried out at a 25 °C. The measurements were repeated 20 times and results were expressed as the mean value of zeta potential ± standard deviation (SD).

### 2.7. Composites Characterisation Methods

#### 2.7.1. X-ray Diffraction (XRD) Analysis

The phase composition of composites was analysed with X-ray diffraction (XRD, D8 ADVANCE from Bruker, Billerica, MA, USA), using a copper sealed tube x-ray source producing Cu Kα radiation at a wavelength λ = 0.154056 nm. Running parameters were as follow: voltage 40 kV, current 40 mA, angular range 20° to 120°, step Δ2Θ − 0.05°, counting time 3 s.

#### 2.7.2. SEM and Transmission Electron Microscopy (TEM) Observations

To confirm the preservation of the MXenes structure and to reveal the effect of Ti/Mo modification on composites mechanical properties, the scanning electron microscopy with an energy-dispersive X-ray spectroscopy (EDS) detector was used (SEM Hitachi S5500, Tokyo, Japan). The microstructure, fracture surface and crack propagation observations were performed. Specimens were coated beforehand with carbon using a sputter (Q150R ES, Quorum Technologies, Lewes, UK). In addition, the interface examinations were conducted with transmission electron microscopy (TEM, TECNAI G2 F20 S-TWIN, FEI, Hillsboro, OR, USA), operating at 200 kV. TEM specimens were prepared by mechanical grinding and subsequent Ar ion polishing at 4 keV to obtain electron-transparent material. In scanning transmission mode (STEM) a Fischione 3000 high-angle annular dark field (HAADF) detector was used to collect images, supplementary to bright field (BF) mode. In order to reveal crystallographic structure of materials, both selected area electron diffraction (SAED), and high resolution TEM (HRTEM) with Fast Fourier transformation (FFT) techniques were adopted.

#### 2.7.3. Mechanical Properties Measurements

The essential properties of the produced composites were thoroughly investigated. Density was measured using a helium pycnometer (Ultrapycnometer 1000, Quantachrome Instruments, Boynton Beach, FL, USA), with 100 tests counted to average. In addition, to evaluate the densification level, sintering curves obtained from the SPS were analysed, showing punch displacement and temperature as a function of process time. The ultrasonic method (refractometer from Optel, Wroclaw, Poland) was used to calculate Young’s modulus. Hardness was measured with the use of Vickers hardness tester (FV-700e, Future-Tech, Kawasaki, Japan). The indentation method (load 98.1 N) was utilized to examine the fracture toughness of composites, calculated from the length of cracks which developed during a test using a Niihara, Morena, Hasselman equation [54]. For each sinter, 7 indentations were conducted, and all of the cracks were marked to be a radial-median system.

## 3. Results and Discussion

The morphology of Ti_3_C_2_ is presented in Figure 1. To confirm the effectiveness of the production method, powder was examined after each technological step. Figure 1a shows the Ti_3_AlC_2_ MAX phase after SPS synthesis and the following grinding process. The close packed, layered structure can be observed, as Al atoms bond together subsequent layers. The synthesised MAX phase consisted dominantly of Ti_3_AlC_2_, along with a small amount of TiC and unreacted graphite, as presented in the XRD results in our previous study [47]. The powder structure after the HF etching process is presented in Figure 1b. As a result of Al atoms’ removal, the subsequent separated Ti_3_C_2_ layers are easily visible. The morphology of the synthesis final product, the MXene phase, is shown in Figure 1c. The carried-out resulted in obtaining few-layered particles of Ti_3_C_2_, characterised by a flake structure with a lateral size of a few microns.

In addition to SEM observations, the phase analysis of synthesised MAX phase and final MXene phase was performed. The obtained diffraction patterns are presented in Figure 2. In the case of MAX phase, diffractogram reveals Ti_3_AlC_2_ as the dominant component, which confirms the effectiveness of the production method. Additionally, some TiC phase as a result of synthesis and small amounts of unreacted titanium can be observed. The diffraction pattern for MXene exhibits signals characteristic for Ti_3_C_2_, which is the dominant phase [55]. This proves that the adopted procedures of etching and delamination have been successful. The presence of titanium oxides was also identified in the diffractogram, which is typical for the Ti_3_C_2_ MXene phase [56].

To confirm the presence of Ti and Mo on the surface of MXene flakes, pristine Ti_3_C_2_ phase as well as those modified with Ti and Mo were investigated using FTIR. The results obtained are presented in Figure 3. The spectrum of pristine Ti_3_C_2_ is similar to our previous work [40]. The characteristic bond vibrations, at 3016 cm^−1^ and 1487 cm^−1^, associated with the presence of C–H, and at 947 cm^−1^ which comes from C–F bonds, can be observed. In the spectrum of the Ti_3_C_2_-Ti, peak at 3011 cm^−1^ derived from C–H vibration is present. A high peak at 2358 cm^−1^, which corresponds to the presence of CO_2_, can also be observed. There is also a peak at 613 cm^−1^ which is assigned as the Ti–O deformation vibration [57,58]. In addition, the peak at 566 cm^−1^ can be attributed to the O–Ti–O lattice stretching vibration of TiO_2_ [59]. The spectrum of the Ti_3_C_2_-Mo is similar to one modified with Ti. The peak at 3034 cm^−1^, derived from C–H vibration, and a high peak at 2361 cm^−1^, which corresponds to the presence of CO_2_, can be observed. The lower side of the wavenumber spectrum shows peaks such as 957 cm^−1^ and 568 cm^−1^, which come from Mo=O and Mo–O bonds [60,61]. In the range of 600–500 cm^−1^ there are peaks associated with the presence of Mo–O vibrations, which confirms the presence of Mo on the surface of the investigated MXene phase [62].

The zeta potential measurements for pristine Ti_3_C_2_ phase as well as that modified with Mo and Ti, in both water and isopropanol, are shown in Figure 4. By analysing these results it can be concluded that the examined phases are more stable in isopropanol than in water, regardless of the modification. This is due to the fact that MXene phases disintegrate in the aquatic environment. Such results were previously presented by other authors, who proved in their research that water, not oxygen, plays a key role in the degradation process of MXenes [63]. The unmodified Ti_3_C_2_ phase in isopropanol is characterised by highest stability, its zeta potential was about −35 mV. Applied modifications resulted in a decrease of zeta potential for Ti/Mo modified MXenes, regardless of the medium. Comparing the specimens with metallic layers, Ti_3_C_2_-Mo showed greater stability than Ti_3_C_2_-Ti, in both water and isopropanol.

The sintering curves obtained for composites with 1 wt.% addition of Ti_3_C_3_/Ti_3_C_2_-Ti/Ti_3_C_2_-Mo and for unreinforced Al_2_O_3_ samples are shown in Figure 5. The recorder shrinkage profile is characteristic for the SPS process, and was previously reported in other studies [64,65]. The displacement curve can be divided into five segments (I-V), separated in the figure with dashed lines. In the first section, specimens are heated from room temperature to the point of the pyrometer working range (250 °C), and no punch movement was noted. Further heating leads to thermal expansion of graphite components (mould and punches), responsible for negative piston displacement in segment II. In the next section, a change in the punch movement direction can be observed, followed by its rapid increase. This results from started compaction, in which takes place such phenomena as particles necking and mass transfer. In the fourth section, the final sintering temperature is reached, and pores present in the material are being sealed. This segment plays the crucial role, as it helps to evaluate the stage of densification. The course of punch displacement is identical for all sinters (pure alumina and composites, regardless the modification), and reaches the plateau zone. This suggests that the sintering parameters were well chosen, and all of the specimens reached full possible densification. It is also important to note that the entire SPS consolidation took only 18 min. Such a fast process should help in preventing the MXene phases from degradation.

The results of density measurements and mechanical properties examinations for all of the specimens are presented in Table 1. High density of 3.93 g/cm^3^, close to theoretical, was achieved for pure alumina sinter. Comparing the composites, series with unmodified Ti_3_C_2_ is characterised by the highest results (from 3.89 to 3.99 g/cm^3^). Similar values, in the range of 3.84–3.94 g/cm^3^, were measured for sinters with Ti and Mo modifications. It is important to note, however, that within each series specimens exhibited the same course of results. The lowest densities were obtained for sinters with 0.5 wt.%, reached maximum for 1 wt.% and decreased for 2 wt.% of phases addition. The Vickers hardness and fracture toughness results obtained for reference specimen (1828 HV10 and 3.98 MPa × m^1/2^), are in line with other studies concerning spark plasma sintered alumina [66]. In the case of Ti_3_C_2_, composites cracked uncontrollably under an indenter creating large areas of cracks and spatters, making mechanical properties measurements impossible to perform (exemplary Vickers indenter figures are presented in the following paragraph). Analysing the series with Ti and Mo, identical results of Vickers hardness were obtained (from 1918 to 2020 HV10), which were slightly higher than for the reference specimen. The course is reversed, however, compared to density measurements. The highest values were obtained for composites with 0.5 wt.% and 2 wt.%, while the lowest for 1 wt.% phase addition. It is important to note that obtained Vickers hardness results are significantly higher (50%) than in other work on similar composites [44]. The fracture toughness results for sinters with Ti_3_C_2_-Ti varies from 3.81 to 4.13 MPa*m^1/2^, and are within the error limit of pure alumina. For composites with molybdenum, the highest fracture toughness of all specimens (4.59 MPa*m^1/2^) was measured for sinter with 0.5 wt.% addition. Further increase of Ti_3_C_2_-Mo content resulted in decrease of K_IC_ (to 3.52 MPa × m^1/2^ for 2 wt.%), which is a phenomenon reported by other authors in composites reinforced with layered crystals [67,68].

The microstructure of composites with Ti_3_C_2_ is presented in Figure 6a–c. In addition to the observations performed, the EDS detector was used to reveal the chemical elements present within the sinters. In Figure 6a, a dense microstructure with fine isometric grains that is free from pores can be seen. Such observations correlate very well with high densities of these sinters. It is important to note that there is no evidence of layered MXene phase within the microstructure. Interestingly, the fracture mode of the specimen is purely intergranular. Except for the ceramic matrix grains, only single other particles were observed. However, they are characterised by a spherical structure as shown in Figure 6b, in contrast to two-dimensional Ti_3_C_2_ used as starting material. EDS analysis revealed a high amount of titanium and carbon for point 1 (corresponding to additional structure), as well as aluminium and oxygen for point 2 (corresponding to alumina grains). Interesting observations were made for the composite with 2 wt.% of Ti_3_C_2_ addition, as presented in Figure 6c. They reveal the presence of a second phase with unidentified structure, and high porosity in the close area. In addition, Figure 6c presents the Vickers indenter imprint for composites with 0.5 wt.% of Ti_3_C_2_. Enormously cracked area with many spatters can be observed, making the mechanical properties’ measurements impossible to perform.

Figure 7a–c presents the microstructure of the composites with 2 wt.% of Ti_3_C_2_-Ti addition. In comparison to sinters with unmodified MXene phase, the one similarity is dense microstructure. On the other hand, the most obvious difference is the change in fracture mode. The composites crack with high participation of transgranular fracture, as shown in Figure 7a. This suggests some significant changes in the microstructure for these specimens. In addition, the flake-shaped structures are visible within the microstructure of composite, as presented in Figure 7b. Their morphology and lateral size correlate well with the synthesised MXene phase incorporated in this study (Figure 1c). The EDS mapping corresponding to particles present in the microstructure in Figure 7c, revealed an area rich in both carbon and titanium. Unlike in the case of composites with unmodified Ti_3_C_2_, mechanical properties examination proceeded normally. The conventional Vickers indenter imprints were obtained, as shown in Figure 7d.

The wt.% of Ti_3_C_2_-Mo, presented in Figure 8a–c, are similar to one of titanium modified phase. Dense microstructures, with no visible pores or voids were obtained. As shown in Figure 8a, also for this material there is a high participation of transgranular fracture mode. This proves that both Ti and Mo modifications resulted in the change of the way the composites cracks. Similar to specimens with Ti modified MXenes, performed observations reveal clearly visible flake-shaped particles with an end twisted and anchored within the matrix (Figure 8b). However, some smaller spherical structures are also present in the microstructure. The EDS mapping performed from such a particle is presented in Figure 8c. It shows carbon- and titanium-rich areas, but with no evidence of significantly higher participation of molybdenum. Similar to sinters with Ti_3_C_2_-Ti, the mechanical properties were fully tested without any obstacles. As presented in Figure 8d, the obtained imprints were well shaped with visible cracks radiating from the Vickers indenter.

In order to determine the character of the presented structures, phase composition analysis was performed. The XRD patterns for composites with 2 wt.% of each of the additions are presented in Figure 9a–c. Except for the Al_2_O_3_ ceramic matrix phase, TiC was identified for each of the specimens, regardless of the addition. It is important to note that typically Ti_3_C_2_ phase undergo an oxidation process following the reaction: Ti_3_C_2_ + 3O_2_ = 3TiO_2_ + 2C [69]. As experimentally proved, this rule applies especially to a water-rich environment, and to a lesser extent to the presence of oxygen [70]. In contrast, the phenomenon of TiC formation has been previously reported for the processes carried out in the vacuum [71]. Authors explained that high temperatures induce rearrangement of Ti and C atoms in the MXene phase, with an outcome of cubic TiC recrystallisation. In the present study, diffractograms for all specimens show peaks corresponding to this phase formation. Interestingly, no presence of TiO_2_ or Ti_3_C_2_ was detected within the diffraction pattern. In addition, some amount of free carbon was detected within composites with Ti (Figure 9b) and Mo (Figure 9c) modified additions. For the latter, a small peak corresponding to molybdenum carbide is also visible. This is consistent with literature reports on Mo_2_C creation [72]. It synthesises at elevated temperatures through carbonisation of MoO_3_ in a carbon rich environment (in this case from MXene atoms rearrangement).

To reveal the nature of the observed structures, specimens were subjected to transmission electron microscopy observations. Figure 10a–d presents TEM analysis for composites with 2 wt.% of unmodified Ti_3_C_2_. Some highly porous areas (with several microns in size) are visible within the microstructure (Figure 10a), previously seen in SEM microstructures. The marked field no. 1 was magnified and further analysed. Figure 10b shows that the individual grains in this porous area are stretched on the other artificial material. In addition, the selected area electron diffraction performed from the pointed spot defines the grain as the Al_2_O_3_. The high-resolution TEM (HRTEM) image with fast Fourier transform pattern from field no. 2 (Figure 10c) reveals the presence of graphite. Similar observations were carried out for other porous areas, as presented in Figure 10d. In this case, the selective area diffraction pattern defines the grain to be TiC, which confirms the XRD results. The HRTEM image performed from marked field no. 3, discloses a good interface between titanium carbide and graphite. Such a good TiC–C bonding is probably an outcome of how these phases emerge. Sintering in a vacuum leads to the rearrangement of Ti and C atoms in MXene phase, as mentioned before [69]. It is important to note that the existence of such porous areas within the entire microstructure may be the reasons for the uncontrolled cracking of the specimens during mechanical properties testing.

The TEM analysis for composites with 2 wt.% of molybdenum modified Ti_3_C_2_ is presented in Figure 11a–d. Some elongated voids, the size of few microns, can be observed in the microstructure, as shown in Figure 11a. Interestingly, bright particles are present at the edges of this void, which correspond to some heavy element (in HAADF mode, the detector shows mass thickness in contrast to signal intensity). The selective area electron diffraction performed from a bigger area obviously confirms the alumina as the matrix grains. To determine the occurring structures, field no. 1 was magnified and analysed. Figure 11b clearly shows an additional phase adjacent to the void. The high-resolution TEM image with fast Fourier transform pattern from field no. 2, reveals the presence of Mo_2_C, confirming the XRD analysis. These findings correlate well with the performed SEM observations. In the case of sinters with Ti/Mo modified phases, flake-shaped particles were present in the microstructure, and their size was equivalent to one of the elongated voids shown in the TEM image. However, the presence of free carbon and the absence of MXene peaks (revealed in XRD patterns), suggest that these voids are rather residues after graphite (than after Ti_3_C_2_ structures), which is being removed during specimen’s preparation. It might be due to weak bonding between metallic layers and initial MXene structures. In this study the Ti/Mo elements were sputtered onto Ti_3_C_2_ phase, thus the character of bonding is purely adhesive. This could be improved with the use of a method providing a chemical reaction, e.g., electroless plating, successfully applied for nickel coated graphene [73]. Similar to sinters with unmodified phase, SADP revealed the TiC particles within the microstructure, as shown in Figure 11c. It is important to note, that in this case there is no porosity at the grain boundaries. In addition, Figure 11d exposes an elongated structure between matrix grain. The HRTEM image with a FFT pattern from field no. 3 confirmed the presence of graphite. These findings suggest that applied modifications changed the way the MXenes degraded during the sintering process. In the case of composites with unmodified Ti_3_C_2_, porous areas consisting of TiC particles stretched on graphite arose. For sinters with modified MXene, except for the TiC phase, there is also the presence of elongated graphitic carbon structures.

## 4. Conclusions

This work presents research on the applicability of MXenes (Ti_3_C_2_) as reinforcing phases in ceramic matrix composites (Al_2_O_3_). In order to prevent the Ti_3_C_2_ from degradation during the sintering process, the structural modifications of sputtering titanium or molybdenum layer on the MXene were adopted. The emphasis of the research was placed on the microstructural analysis and mechanical properties measurements. The observations performed suggest that the applied modifications changed the way the MXenes degraded during the sintering process. In the case of composites with unmodified Ti_3_C_2_, porous areas consisting of TiC particles stretched on graphite arose. For sinters with modified MXene, except for TiC phase, there is also a presence of elongated graphitic carbon structures. It is expected that the metallic layer created at the Ti_3_C_2_ phase works as a barrier for the migration and rearrangement of atoms. In addition, it can limit the diffusion with the environment and consequently change the degradation process. In addition, high mechanical properties were measured for composites with Ti_3_C_2_-Ti and Ti_3_C_2_-Mo. For both of these modifications, the Vickers hardness results show improvement within the entire range of addition, and the highest measured values (2017 and 2020 HV10), were 10% higher when compared to reference specimen. As for the fracture toughness, an increase of 15% was achieved for the specimen with 0.5 wt.% Ti_3_C_2_-Mo, while results for series with Ti_3_C_2_-Ti were within the error limit for pure alumina. An increase of mechanical properties (for sinters with Ti_3_C_2_-Ti/Mo), can be attributed to the in situ creation of graphitic carbon (which will work as flake-shaped reinforcement), and to the non-porous interface at the phase boundaries. In the case of sinters with unmodified phase, the mechanical properties’ examinations were impossible to perform due to uncontrolled cracking with enormously ruptured areas. Moreover, the fracture changed from purely intergranular to cracking with high participation of transgranular mode, analogously. Further research should focus on applying chemical reaction methods in order to ensure a better interface between modification and MXenes.

## Figures and Tables

**Figure 1 materials-14-00829-f001:**
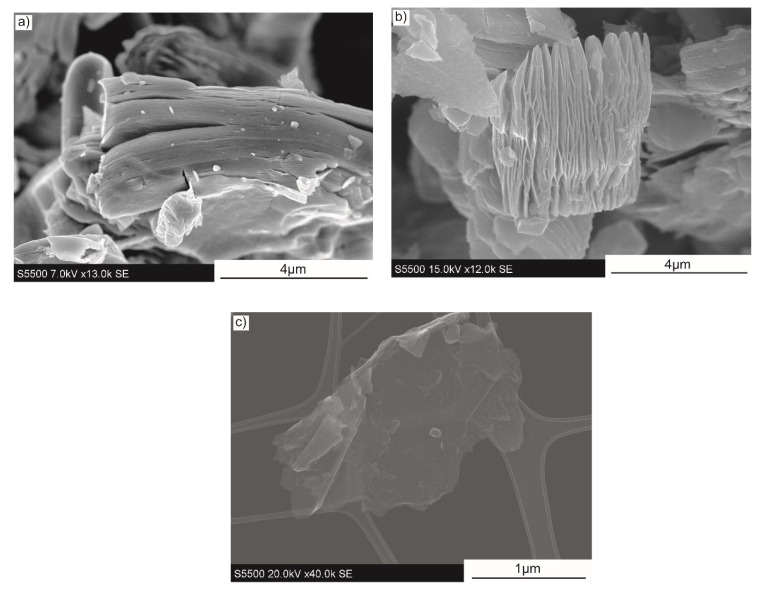
The morphology of (**a**) synthesised Ti_3_AlC_2_, (**b**) etched MAX phase, (**c**) Ti_3_C_2_ MXene.

**Figure 2 materials-14-00829-f002:**
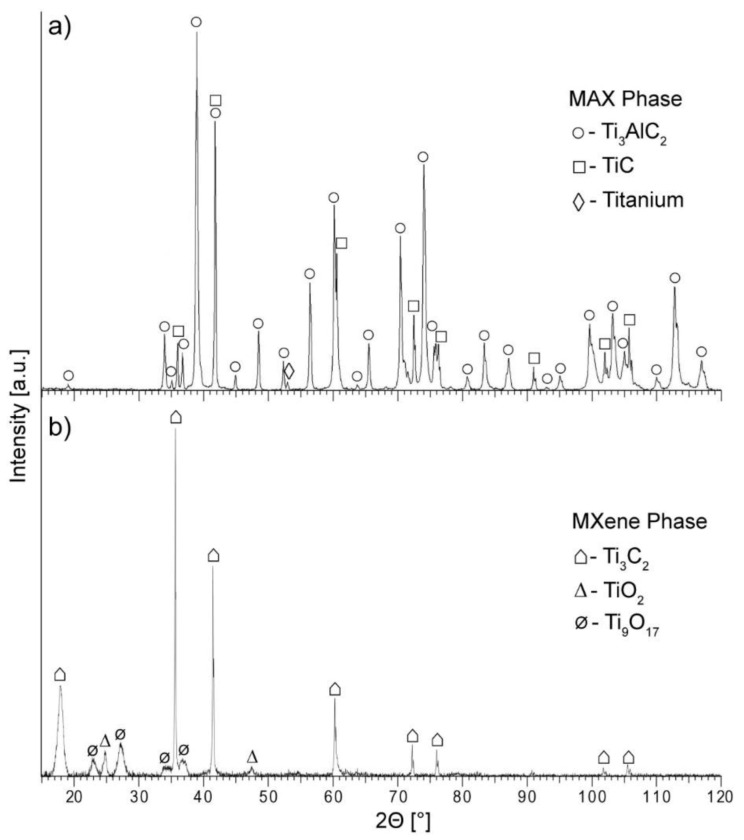
X-ray diffraction (XRD) patterns for (**a**) synthesised Ti_3_AlC_2_ MAX phase, (**b**) Ti_3_C_2_ MXene phase.

**Figure 3 materials-14-00829-f003:**
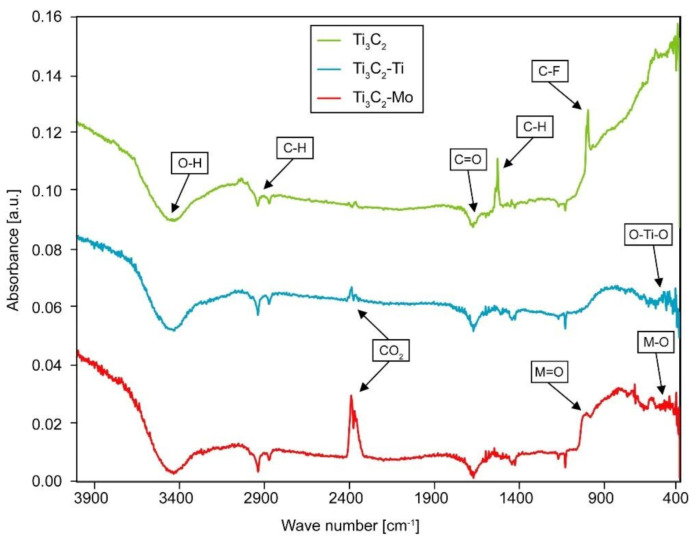
Fourier transform infrared spectroscopy (FTIR) analysis of pristine Ti_3_C_2_, Ti_3_C_2_-Ti and Ti_3_C_2_-Mo flakes.

**Figure 4 materials-14-00829-f004:**
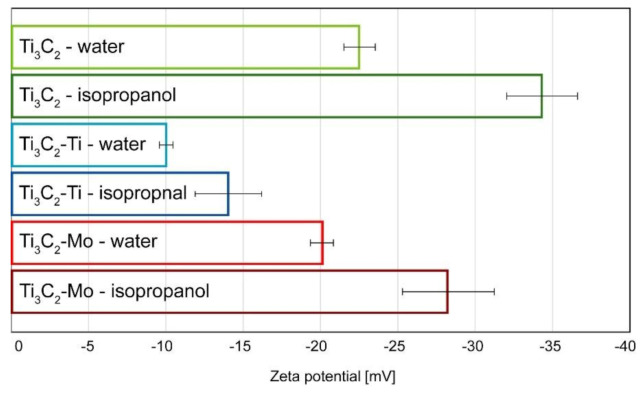
Zeta potential of pristine Ti_3_C_2_, Ti_3_C_2_-Ti and Ti_3_C_2_-Mo in water and in isopropanol.

**Figure 5 materials-14-00829-f005:**
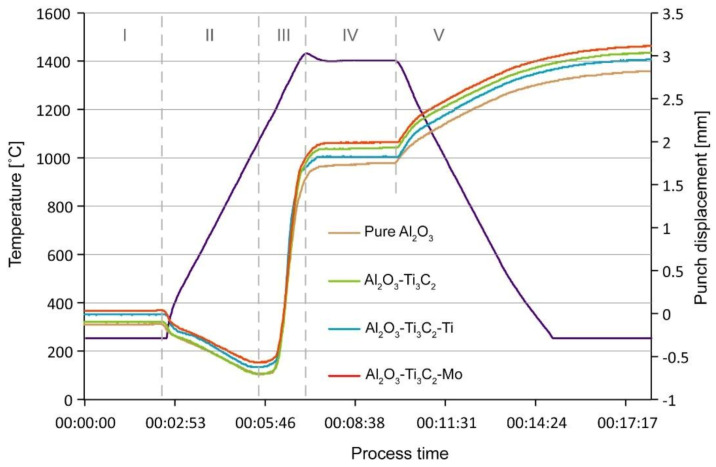
Spark plasma sintering (SPS) curves of Al_2_O_3_-1 wt.% Ti_3_C_3_/Ti_3_C_2_-Ti/Ti_3_C_2_-Mo composites and pure Al_2_O_3_ specimen.

**Figure 6 materials-14-00829-f006:**
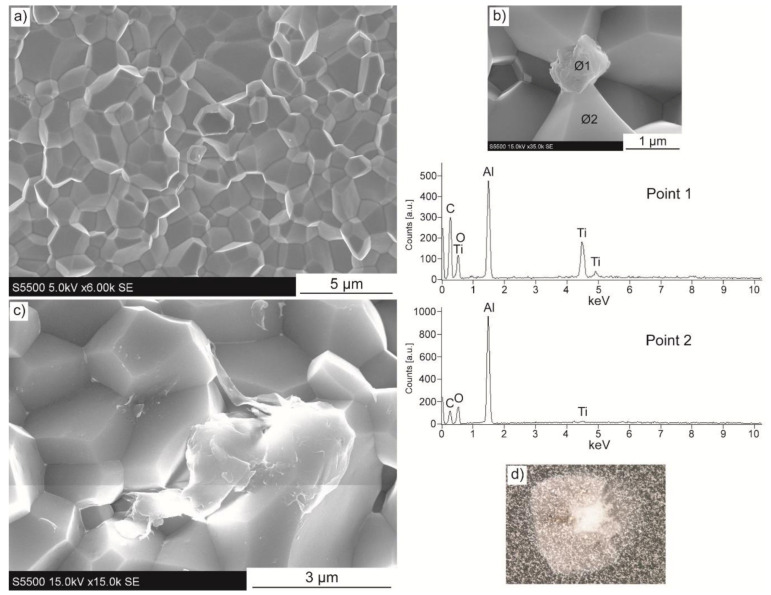
Microstructure images of composites with Ti_3_C_2_ addition of (**a**) 1 wt.%, (**b**) 1 wt.% with energy-dispersive X-ray spectroscopy (EDS) points analysis, (**c**) 2 wt.%, (**d**) 0.5 wt.% with Vickers indenter imprint.

**Figure 7 materials-14-00829-f007:**
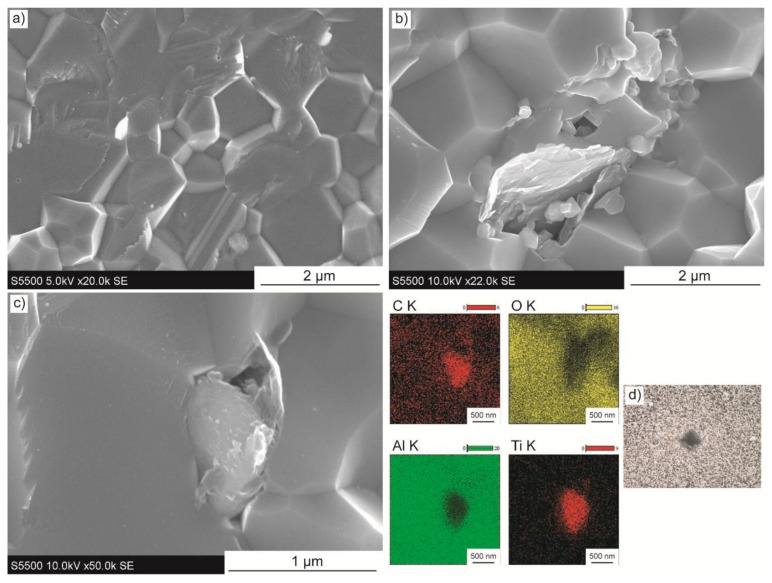
Microstructure images of composites with Ti_3_C_2_-Ti addition of (**a**,**b**) 2 wt.%, (**c**) 2 wt.% with EDS map of elements, (**d**) 0.5 wt.% with Vickers indenter imprint

**Figure 8 materials-14-00829-f008:**
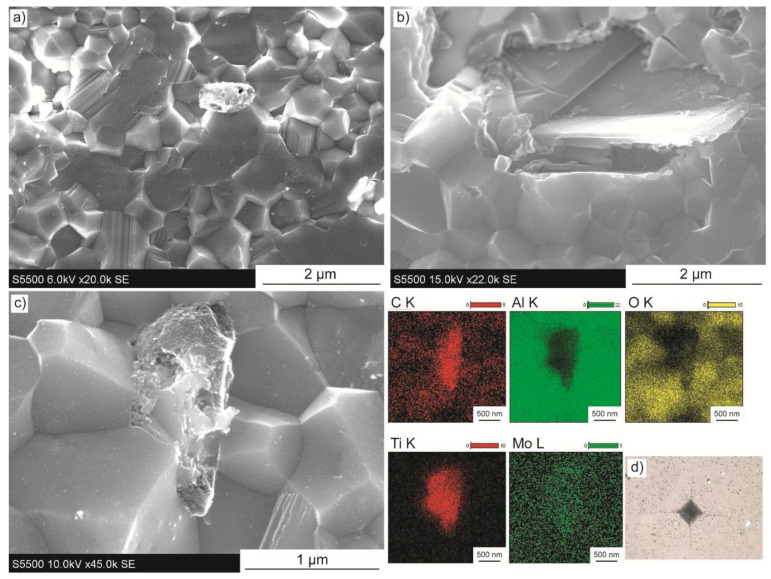
Microstructure images of composites with Ti_3_C_2_-Mo addition of (**a**,**b**) 1 wt.%, (**c**) 2 wt.% with EDS map of elements, (**d**) 0.5 wt.% with Vickers indenter imprint.

**Figure 9 materials-14-00829-f009:**
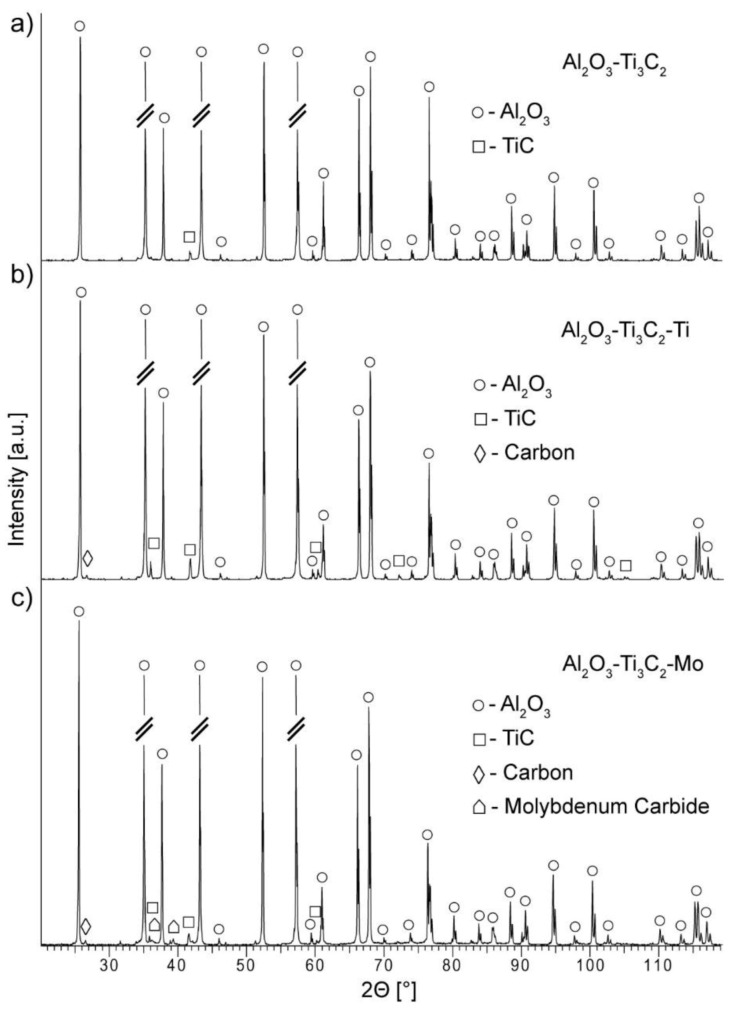
XRD patterns for composites with 2 wt.% of (**a**) Ti_3_C_2_, (**b**) Ti_3_C_2_-Ti, (**c**) Ti_3_C_2_-Mo.

**Figure 10 materials-14-00829-f010:**
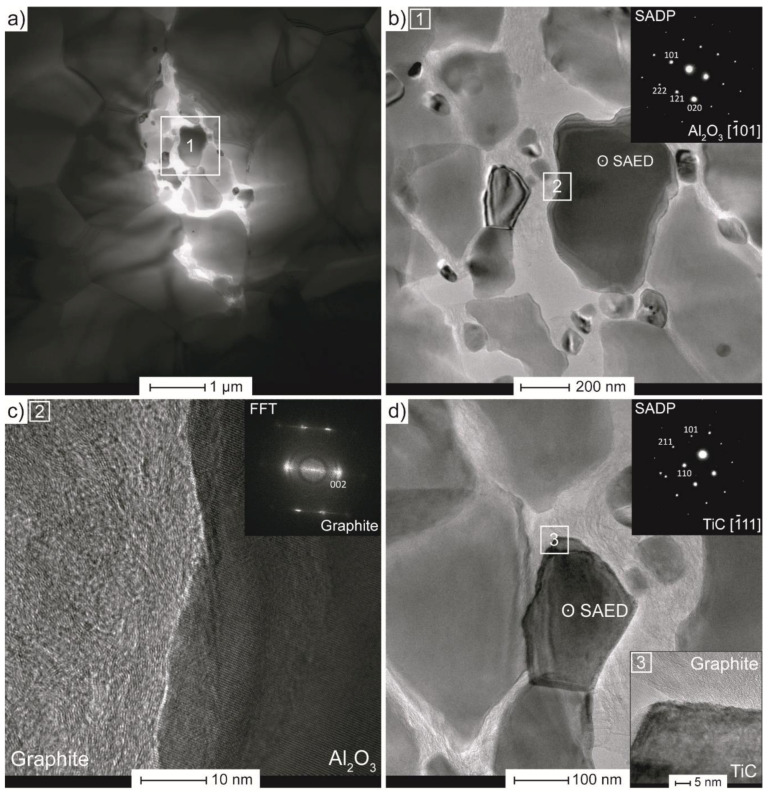
Transmission electron microscopy (TEM) analysis of Al_2_O_3_ with 2 wt.% of Ti_3_C_2_: (**a**) microstructure in bright field (BF) mode, (**b**) magnified image from porous area with selected area diffraction pattern (SADP), (**c**) high-resolution TEM (HRTEM) image with Fast Fourier transform (FFT) pattern, (**d**) second phase with SADP and HRTEM image showing its grain boundary.

**Figure 11 materials-14-00829-f011:**
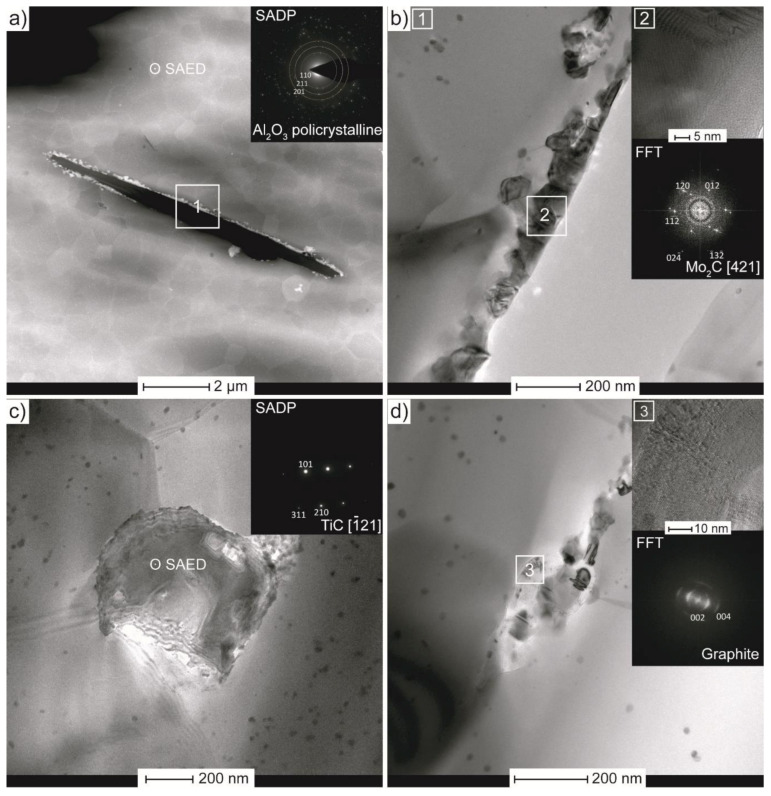
TEM analysis of Al_2_O_3_ with 2 wt.% of Ti_3_C_2_-Mo: (**a**) microstructure in high-angle annular dark field (HAADF) scanning transmission mode (STEM) mode with SADP, (**b**) magnified image from elongated void with HRTEM and FFT pattern, (**c**) second phase located between alumina grains with SADP, (**d**) artificial phase at grain boundary with HRTEM and FFT pattern.

**Table 1 materials-14-00829-t001:** Sinters density and mechanical properties.

Specimen		Density (g/cm^3^)	Hardness (HV10)	K_IC_ (MPa*m^1/2^)
**Pure Al_2_O_3_**		3.932 ± 0.32	1828 ± 23	3.98 ± 0.32
Al_2_O_3_ + x% Ti_3_C_2_	0.5	3.889 ± 0.002	N/A	N/A
	1	3.999 ± 0.004	N/A	N/A
	2	3.918 ± 0.003	N/A	N/A
Al_2_O_3_ + x% Ti_3_C_2_-Ti	0.5	3.848 ± 0.007	1983 ± 24	4.06 ± 0.31
	1	3.947 ± 0.004	1918 ± 57	3.81 ± 0.17
	2	3.858 ± 0.008	2017 ± 44	4.13 ± 0.32
Al_2_O_3_ + x% Ti_3_C_2_-Mo	0.5	3.855 ± 0.005	2020 ± 31	4.59 ± 0.35
	1	3.900 ± 0.004	1953 ± 31	3.74 ± 0.28
	2	3.877 ± 0.007	2000 ± 26	3.52 ± 0.25

## Data Availability

Not applicable.

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
