# Peer review of "Microstructure and Mechanical Properties of Alumina Composites with Addition of Structurally Modified 2D Ti3C2 (MXene) Phase"

_materials, 2021, doi:10.3390/ma14040829_

Round 1
Reviewer 1 Report
The paper ‘Microstructure and mechanical properties of alumina composites with addition of structurally modified 2D Ti3C2 (MXene) phase’ presents a thorough study of the fabrication and characterization of MXenes as reinforcing phases in ceramic matrix composites. This paper is of interest for the field. Suggested corrections are listed below:
(1) The paper should be revised by a native speaker.
- The article ‘the’ is missing at many places. Examples are given below:
- Line 86: Authors report to
- Line 94: as pointed out by authors
- Line 359: Authors explained
- Line 165: Composites presented in this study
- Line 289: Lowest density
- Line 317: for composite with 0.5 wt% of Ti3C2
- Some sentences need to be rewritten. Some examples are:
- Abstract : high results of mechanical properties were obtained for composites
- Line 76: To date, there are only single articles in the literature regarding
- Line 309: At Figure 5a, free from pores, dense microstructure with fine isometric grains…
- Line 166. Replace ‘In the first technological step’ by ‘First,’
(2) In the abstract: It would be helpful to add the % of improvement in the Vickers hardness and fracture toughness.
(3) Section 2.2: Was a graphite die used for the SPS? How do the authors get rid of the superficial carbon after sintering?
(4) Table 1: why aren’t there uncertainties for the samples Al2O3 + x% Ti3C2?
(5) Figure 5: Increase font of the text in the EDX spectra, and also the font for point 1 and 2 on the micrograph 5b
(6) Figures 6 and 7: Increase font in the EDX map
(7) Figure 8: It would be easier to compare if the three patterns were plotted on the same graph, with a vertical shift.
(8) Line 362: the authors used the word spectra for the XRD data. This is not correct as the energy does not vary. Replace spectra by diffraction pattern or diffractogram
(9) Why not showing TEM of Al2O3 with 2 wt% of Ti3C2-Ti?
(10) In the conclusions, the authors do not mention the improvement of the mechanical properties. Adding a sentence with the % of improvement in the Vickers hardness and fracture toughness is recommended
Reviewer 2 Report
The experimantal data obtained by the authors are new. The paper is interesting and fits joural profile.
Comments:
- The characterization of MAXene phase is incomplete. The SEM data presented in Fig. 1 for both MAX phase and MAXene are very similar. It should be desirable to add X-ray diffractio patterns of both initial MAX pahse and MAXene obtained after Al leaching.
- It is not clear, what composition was formed after the procedure of surface treatment of MAXene with Ti and Mo? Indeed, the mass of MAXene loaded in the reactor was 0.5 g, The volume of sputtered Ti and Mo targets were 0.527/0.577cm3. Therefore, the estimated values of mass of Ti and Mo in the final mixtures should be 2.37/5.91 g, respectively, i.e. the concentration of the metal additive should be very large. Authors should point a strict concentration of metals in the initial MaXene-metal powders.
- FTIR spectra seem to be very strange: adsorption bands for Đž-H, C-H, C=O vibrations has a true shape on the spectrum (I/e/ corresponds to light adsorption), whereas CO2 (fragments?) , C-H (at ~1500 cm-1), C-F, M=O vibrations bands correspond to the IR tramsmission spectrum rather than the IR adsorption spectrum. What is a reason of appearance of such a strange bands on the spectrum?
The paper may be published only after the revision in accord with the above comments.
Reviewer 3 Report
The present work addresses the study of the microstructure and the mechanical properties of alumina composites with MXenes phases. The synthesis of the MAX phase, the obtaining and structural modifications of the MXene phases and the composites fabrication by Spark Plasma Sintering were also accomplished by the authors.
I think the topic of the paper is interesting and original, and the experimental design is correct, however, I have some concerns and questions that should be amended before publishing the paper.
My main concern is that the authors conclude that it is possible to prepare ceramic composites while preserving the structure of MXene phase (page 15, lines 434-435) and this is not supported by the experimental results at all. The XRD patterns showed no presence of Ti3C2 phase in any of the composites. The detected phases were TiC, Mo2C and graphite. This was supported by the TEM study, which also found graphite, TiC and Mo2C, and definitely absence of MXene phases. I agree with the authors that the introduction of Ti and Mo on the MXene phase induces significant changes in the MXene degradation process, but in the three different studied composites the MXene phases were completely degraded. Thus, this conclusion (page 15, lines 434-435) should be removed.
Also, the authors claim for an enhancement of some mechanical properties in the composites with Ti/Mo modified MXenes (page 8). However, these phases are not really present in the composites, so what would be the reason for the enhancement of the properties?
The XRD, EDS and TEM analysis revealed the presence of spherical TiC particles in the composites. The presence of this phase is related to the degradation of Ti3C2 during processes carried out in vacuum (page 11). What is the reason for the change of shape in these particles, from the initial 2D morphology for the synthesized MXene phase to the spherical shape in the TiC particles?
In Figure 9 (TEM analysis of Al2O3/Ti3C2 composite) some highly porous areas are revealed, that were not observed in the SEM image presented in figure 5. The authors pointed out that this could be the result of uncontrolled cracking of the specimens during mechanical properties testing. Did the authors perform TEM analysis on this composite previous to the mechanical characterization? This analysis would reveal the true porosity state of the composite.
How can the authors explain the different ways of degradation in the unmodified/modified MXene phases?
Figures 5, 6, 7 and 8 are not properly presented. Numbers, scales, insets and axis titles are impossible to see.
Round 2
Reviewer 3 Report
After the revisions, the paper can be published.